# Emergent coherent modes in nonlinear magnonic waveguides detected at ultrahigh frequency resolution

K. An [1,2,4], M. Xu [1,4], A. Mucchietto[1], C. Kim [2], K.-W. Moon [2], C. Hwang [2] & D. Grundler [1,3] ✉

Nonlinearity of dynamic systems plays a key role in neuromorphic computing, which is expected to reduce the ever-increasing power consumption of machine learning and artificial intelligence applications. For spin waves (magnons), nonlinearity combined with phase coherence is the basis of phenomena like Bose–Einstein condensation, frequency combs, and pattern recognition in neuromorphic computing. Yet, the broadband electrical detection of these phenomena with high-frequency resolution remains a challenge. Here, we demonstrate the generation and detection of phase-coherent nonlinear magnons in an all-electrical GHz probe station based on coplanar waveguides connected to a vector network analyzer which we operate in a frequency-offset mode. Making use of an unprecedented frequency resolution, we resolve the nonlocal emergence of a fine structure of propagating nonlinear magnons, which sensitively depends on both power and a magnetic field. These magnons are shown to maintain coherency with the microwave source while propagating over macroscopic distances. We propose a multi-band four-magnon scattering scheme that is in agreement with the field-dependent characteristics of coherent nonlocal signals in the nonlinear excitation regime. Our findings are key to enable the seamless integration of nonlinear magnon processes into high-speed microwave electronics and to advance phase-encoded information processing in magnonic neuronal networks.

Nonlinear systems have drawn significant attention in recent years due to their importance in fields such as deep learning and artificial neural networks, where nonlinearity is essential for model training[1–4]. Recently, magnetic systems have been proposed as a promising platform for implementing neural networks and nanometric spin-based logic devices considering their low energy consumption and nonlinear behavior[5–11]. When microwave signals excite magnetic materials with high intensity, a diverse range of nonlinear processes can take place. Among them, there are frequency conversion via parametric pumping[12], the Suhl instability[13] and scattering with other quasiparticles[14]. Nonlinear frequency shifts gave rise to a directional coupler holding promise for an integrated magnonic half-adder[15]. Meanwhile, the underlying relaxation processes allowed one to access magnons with diverse wave vectors **k** that can be used for magnon-based computing[16], a reservoir-computing implementation[17], neuro-morphic computing[18], and pattern recognition[19]. In all these works,

[1]Laboratory of Nanoscale Magnetic Materials and Magnonics, Institute of Materials (IMX), School of Engineering, École Polytechnique Fédérale de Lausanne (EPFL), Lausanne 1015, Switzerland. [2]Quantum Technology Institute, Korea Research Institute of Standards and Science, Daejeon 34113, Republic of Korea. [3]Institute of Electrical and Micro Engineering, School of Engineering, École Polytechnique Fédérale de Lausanne (EPFL), Lausanne 1015, Switzerland. [4]These authors contributed equally: K. An, M. Xu. ✉e-mail: dirk.grundler@epfl.ch

the phase coherency of nonlinear magnons was not explored, although the phase information is crucial for signal reconstruction, synchronization, and modulation.

To advance the usage of nonlinear magnons, phase-sensitive electronics are urgently needed which allow for the detection of the creation and evolution of phase-stable nonlinear magnonic responses. Recently, super-Nyquist sampling magneto-optical Kerr microscopy was introduced to perform direct phase-resolved imaging of nonlinear spin waves in a Permalloy ellipse[9]. A specific lateral size and magnetic field were required to suppress stochastic switching between different magnon states[20] and arrive at a standing wave pattern. Only by seeding the magnetic system with an additional frequency component of crucially adjusted power and phase, the nonlinear standing spin waves were detected optically with a frequency resolution of a few MHz[21]. The technique did not detect propagating spin waves.

Parametric pumping is a powerful technique for inducing nonlinear interactions among propagating magnons[12,13]. In this process, an incoming microwave photon excites two counter-propagating half-frequency magnons with finite wave vectors, each carrying half the energy of the photon[22]. The threshold behavior of parametric excitation has been extensively studied[23,24], and the frequency comb characteristics have drawn significant attention recently[25–27]. The importance of phase in parametric pumping has been highlighted in prior studies[24,28–33], but electrical detection schemes based on delay lines have not explored the phase coherency of different nonlinear modes so far[34,35]. The limited frequency resolution of inelastic light

scattering (Brillouin light scattering, BLS)[36,37] is not adequate for capturing magnon dynamics in the kHz to MHz range. A major step in functional nonlinear magnonics would be reached by an all-electrical scheme that addresses both stationary and propagating nonlinear spin waves in a phase-sensitive manner. Such a scheme paves the way for on-chip information processing by means of nonlinear spin-wave processes.

Here, we present a method for precisely measuring the frequency structure of parametrically excited magnons which undergo nonlinear scattering while maintaining their coherency with the microwave source over macroscopic distance. The process of parametric pumping preserves a specific phase relationship between the source and the parametrically excited waves[38–40]. Making use of a frequency offset mode in a vector network analyzer (Fig. 1a), we detected phase-sensitively nonlinear magnons propagating over a macroscopic distance $d$ between two coplanar waveguides (CPWs). We exploited the parallel pumping geometry and detected characteristic fine structures in magnon spectra with a high-frequency resolution of kHz. We attribute them to four-magnon scattering among closely situated magnon bands. Four-magnon scattering allows magnons to occupy frequency positions that satisfy stringent energy and momentum conservation laws. Our frequency-offset configuration provides an unprecedented detection scheme for magnon frequency combs, which can be fine-tuned via magnet shape, microwave power, and field. Unlike the previously applied methods involving photon counting of laser light, our approach ensures seamless electronic integration and promises fast

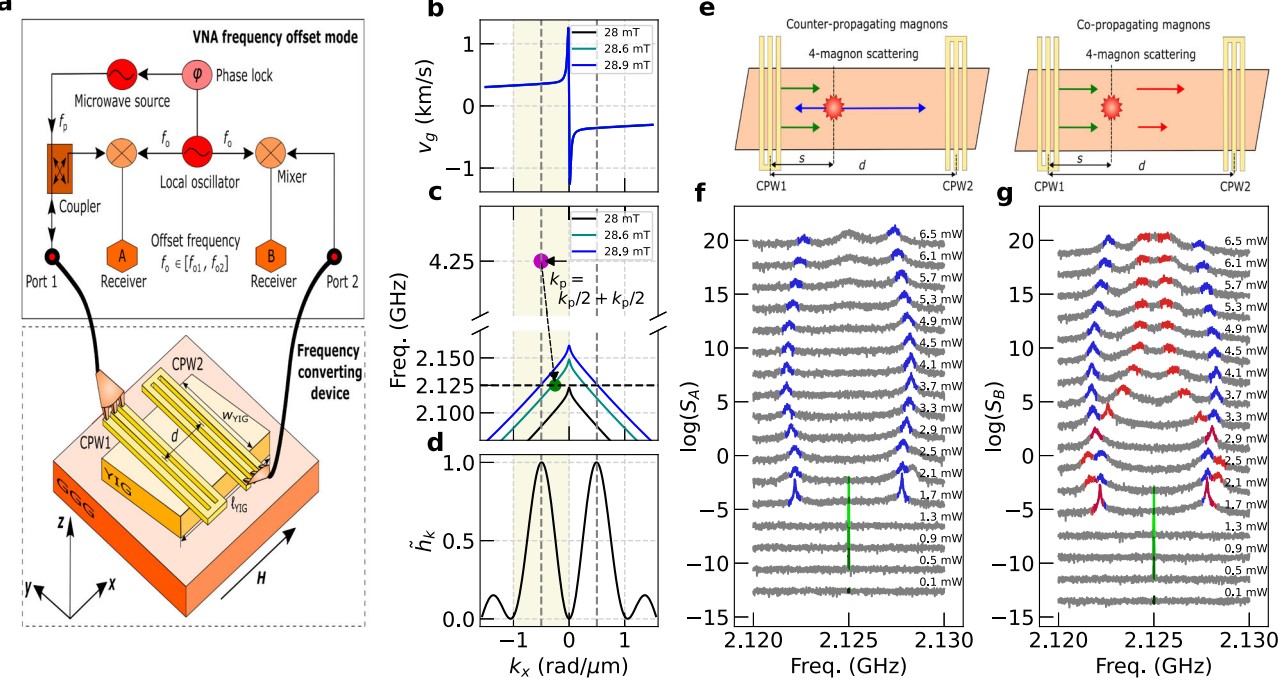

**Fig. 1 | VNA frequency offset measurements of emergent nonlinear spin waves. a** Schematics of the frequency offset measurement implemented on a VNA attached to two CPWs on YIG. The RF signal of the microwave source operated at frequency $f_p$ is applied to CPW1. The local oscillator operates at the offset frequency $f_o$ swept between $f_{o1}$ and $f_{o2}$. Mixed signals from the local oscillator and the nonlinear magnons are detected via two Receivers A and B, which detect phase-coherently voltage signals at CPW1 and CPW2, respectively. **b** Wave-vector dependent group velocity $v_g$ at 28, 28.6, and 28.9 mT. No difference is visible at this scale. **c** Dispersion relation for BVMSWs illustrating the nonadiabatic parametric pumping via forced spin precession at $f_p$ and $k_p$. **d** Fourier analysis of the $x$-component of the normalized RF magnetic field $\tilde{h}_k = h_k / h_k^{max}$. **e** Four-magnon scattering processes outside CPW1. The green arrows indicate magnons created by the nonadiabatic pumping process, which have equal wave vectors and frequencies $k_p/2$

and $f_p/2$. At further increased power, they undergo two different processes with time delays that lead to the scattering center outside CPW1 by $s$. The blue arrows indicate the resultant counter-propagating magnons after the four-magnon scattering. These are detected from both **f** Receiver A (CPW1) and **g** Receiver B (CPW2) represented by the blue-colored peaks. Another type of scattering results in two co-propagating magnons, which can only be detected in CPW2 as they propagate in the direction of their parent magnons (red-colored peaks in **g**). **f, g** Waterfall plots of scattering parameters $S_A$ (signal taken from CPW1) and $S_B$ (signal taken from CPW2) measured for different powers. At low powers, a very sharp peak at $f_p/2$ appears, which has a linewidth on the order of tens of kHz, close to our frequency resolution (green colors in **f** and **g**). At a power of 1.7 mW, it splits into a pair of symmetric side peaks with a linewidth of sub-MHz. The in-plane magnetic field of 28.6 mT was applied along the $x$-direction.

data acquisition in neuromorphic computing by an all-electrical scheme. The long-distance coherency of nonlinear magnons, which we discover in our YIG device, is consistent with a recent work by Makiuchi et al.[41]. While they investigated temporal coherency in a single isolated YIG disk, we investigate here the spatial coherency of propagating magnons encoded in multiple different frequency components. Our findings enable the all-electrical processing of coherent nonlinear signals in magnonic waveguides.

## Results

### Coherent detection of emergent nonlinear spin waves

Our experimental setup (Fig. 1a) was based on a vector network analyzer (VNA) whose two ports were connected to two CPWs. Both CPW1 and CPW2 were employed to excite and detect propagating magnon modes. The respective intensities from CPW1 and CPW2 were recorded in Receiver A ($S_A$) and Receiver B ($S_B$) of the VNA. The implemented frequency offset function enabled the detection of voltage signals at frequencies $f_o$ that deviated from the source port frequency $f_p$. The offset-frequency range ($[f_{o1}, f_{o2}]$) for both CPW1 and CPW2 was between $f_{o1} = f_p/2 - \delta = 2.12$ GHz and $f_{o2} = f_p/2 + \delta = 2.13$ GHz while maintaining the source frequency constant at $f_p = 4.25$ GHz. In this work we chose increments of 7 kHz to vary $f_o$. We note that a seeding signal was not needed and increments could be reduced if required. The power of the RF source was adjusted from 0.1 mW to 6.5 mW.

The frequency conversion was explored on a device prepared from a 130-nm-thick yttrium iron garnet (YIG) grown by liquid phase epitaxy on a 0.5-mm-thick gadolinium gallium garnet (GGG) substrate[42]. The YIG film was shaped into a parallelogram via ion-beam etching, with the sides angled at approximately 10° to reduce edge reflections. The lateral dimensions of the YIG devices were $w_{YIG} = 173$ μm and $\ell_{YIG} = 261$ μm. The allowed magnon bands of the YIG were studied in ref. 43. The devices situated within an external magnetic field **H** maintained along the $x$-direction (Fig. 1a) such that we addressed backward volume magnetostatic spin waves (BVMSWs). Expected group velocities $v_g$ and dispersion relations $f(k_x)$ for different field strengths are shown in Fig. 1b, c. **k** is the in-plane wave vector.

The CPW signal and ground lines had a width of 3.3 μm, and the edge-to-edge spacing between signal and ground lines measured 2.7 μm. The center-to-center distance $d$ separating two parallel CPWs was 30 μm. The CPWs produced a pronounced torque via the $x$-component of the RF field $h_x$ underneath CPW1. Its Fourier analysis shown in Fig. 1d provided the wave vectors $k_x$ relevant for the excitation of BVMSWs in YIG. A large signal is expected for $k_x = k_{CPW} = -0.5$ rad/μm, where the negative sign indicates the opposite direction of $v_g = 2\pi(df/dk_x)$ (Fig. 1b). Note that the efficiency of detection at CPW2 is also $k$-dependent and follows the same wave-vector distribution, i.e., the strongest signals appear for $k_x$ around $k_{CPW}$ assuming a $k$-independent decay length. Therefore, the most effectively excited spin-precessional motion has a wavelength $\lambda_{CPW} = 2\pi/k_{CPW}$, which is about 12 μm. This value of $\lambda_{CPW}$ is larger than the width of the signal line of 3.3 μm across which the CPW provides the largest torque for excitation. This leads to the so-called nonadiabatic condition, where the pumping area is smaller than the wavelength of the spin wave.

The theoretical approach of ref. 44 considers parametric pumping in the case of an inhomogeneous RF magnetic field $h_x$ in BVMSW geometry. In this nonadiabatic condition, the pumping field is represented in $k$ space as the sum of monochromatic pumping waves exhibiting $k$-dependent amplitudes $h_k$. In our experiment, $h_k$ is significant for $-2k_{CPW} < k_p < 0$ (yellow shaded region in Fig. 1b–d). Because of the strict momentum and energy conservation laws requiring $f_p = f_0 + f_0$ and $k_p = k_0 + k_0$, the magnetic field range where co-propagating waves are generated is limited. In Fig. 1c, we plot magnon dispersion relations for three different fields. Below 28 mT, no spin waves exist at $f_p/2$. Larger than 28.9 mT, the spin waves with $k_p < -1$ rad/μm should participate to excite $k_0 = k_p/2$, which has very low efficiency.

Therefore the nonadiabatic process has a limited field range of about 1 mT in our configuration for the magnon bands considered here. The authors of ref. 44 noted that, in spite of the pumping smearing in $k$ space, spin waves remain spectrally narrow in the frequency space similar to adiabatic pumping due to the pronounced frequency selectivity of the parametric interaction.

We first highlight exemplary spectra $S_A$ taken at 28.6 mT (Fig. 1f, g). Increasing the power at port 1 of the VNA beyond 0.5 mW, we see the emergence of a sharp peak at $f_p/2$ (green colors). Its linewidth (full width at half maximum) is tens of kHz covering only several data points (see Supplementary Fig. S5 for a closer view of the spectrum). The same low threshold power and sharp feature at $f_p/2$ are found in the spectra taken on both CPW1 (local) and CPW2 (nonlocal) in Fig. 1f and g, respectively. These two datasets are striking in that first, the parametrically excited spin waves with the lowest threshold power possess surprisingly narrow linewidths confirming the strong selectivity of the nonlinear scattering process, as noted in ref. 44. The sharp peaks confirm that the magnon creation process is governed by the very selective energy and momentum conservation laws and the linewidth does not reflect the magnon damping. Second, the phase-sensitive voltage signals detected at Receiver B of the VNA in Fig. 1g indicate that the nonlinear spin waves remain phase coherent after propagating over a macroscopic distance $d$ of 30 μm and maintain their surprisingly small linewidth.

An increased power $P$ leads to the broadening of the $f_p/2$ peak. Then, at $P \geq 1.7$ mW, a pair of side peaks (marked in blue in Fig. 1f, g) emerges in both the local and nonlocal signals obtained on CPW1 and CPW2, respectively. The side peaks exhibit intensity asymmetry. The peak with stronger intensity resides at $f > f_p/2$ ($f < f_p/2$) in CPW1 (CPW2). These peaks are broader than the one of magnons excited parametrically at $f = f_p/2$ at the threshold power (see Supplementary Fig. S6 for the power-dependent linewidth evolution).

Remarkably, we observe another type of signal in Fig. 1g detected nonlocally in Receiver B only. These peaks (marked in red color) show a distinct power dependence, which indicates that their origin is clearly different from the previously discussed peaks (marked in blue color) as they do not appear in both CPWs. We note that in CPW2, we detect propagating spin waves whose wave vectors **k** exhibit a non-zero $x$-component and whose decay lengths (group velocity $v_g$ divided by relaxation rate) are large compared to $d$. Considering the field **H** applied along the $x$-direction, such spin waves exhibit the character of a BVMSW.

Next we discuss two different four-magnon scattering scenarios following the parametrically excited co-propagating spin waves. The initial spin waves with $f_p/2$ and $k_p/2$ are depicted in Fig. 1e as green arrows. At large pumping powers ($P \geq 1.7$ mW), they can undergo a four-magnon scattering process, which leads to a further pair of magnons with different wave vectors and frequencies fulfilling the conservation laws. It has been established that a certain time $\tau_s \approx 15$ ns after the start of pumping is required to establish the maximum scattering probability[45]. During this period, the initial spin waves propagate a distance $s = v_g \times \tau_s \approx 15$ ns $\times$ 400 m/s = 6 μm ($v_g \approx 400$ m/s at $k_x = 0.25$ rad/μm, see Fig. 1b) along the $x$-direction. As a consequence, the highest probability for four-magnon scattering processes exists outside the CPW1 area (see red burst symbol in Fig. 1e).

We distinguish two scattering scenarios in the following. In one case, they generate counter-propagating spin waves (blue arrows in the left panel of Fig. 1e). In the second case, they generate co-propagating ones (red arrows in the right panel of Fig. 1e). Spin waves depicted with the blue arrows can be detected by both CPWs. However, due to the momentum conservation, the co-propagating spin waves represented by the red arrows can be picked up only by CPW2. The additional peaks observed in CPW2 (red-colored features in Fig. 1g) are consistent with the latter scattering process. This explains the rich spectra with multiple peaks in Receiver B (CPW2). They are

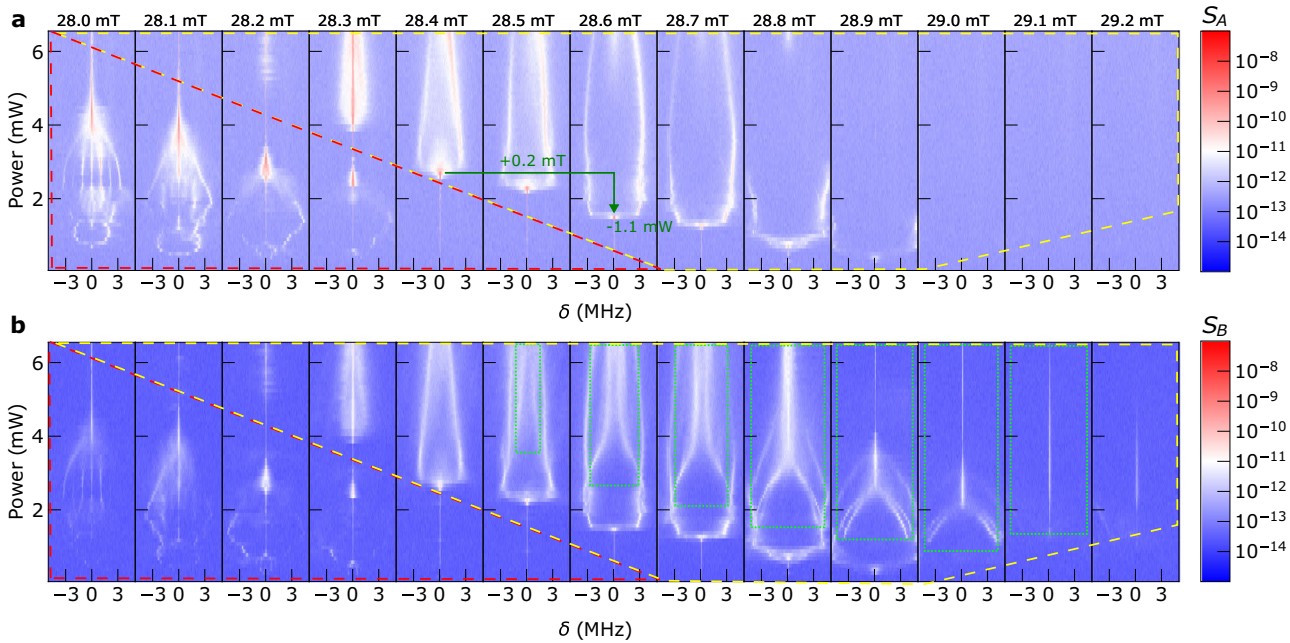

**Fig. 2 | Frequency anatomy of nonlinear spin waves.** Spectra taken at different powers $P$ with a frequency resolution of 7 kHz near $f_p/2$ within a regime of $\delta \equiv f_o - 2125$ MHz = ±5 MHz on **a** CPW1 and **b** CPW2 for increasing magnetic field (from left to right). The spectral region covered by a red dashed triangle represents the low field regime. It features the formation of a magnon frequency comb (MFC)-like structure, where the frequency spacing changes with field and power. The MFC-like feature merges into a $f_p/2$ when increasing power. In a high field regime (represented by the yellow dashed-line enclosure in the upper right area), the $f_p/2$ splits into a pair of side peaks, where the point of splitting shifts toward lower power with increasing field. The amount of shift establishes the power-field relation of about −5.5 mW/mT (see the green arrows in **a**). Between 28.5 and 29.1 mT, the signals obtained on CPW2 (green dotted rectangles) exhibit significantly richer features than on CPW1, indicating that scattering processes and the occupation of further magnon states take place outside CPW1.

absent in Fig. 1f. Note that the scattering processes depicted in Fig. 1e take place on both sides of CPW1 as $k_p$ can be both positive and negative. For scattering processes on the left side of CPW1, again the blue-colored one produces spin waves detected by both CPW1 and CPW2. For the red-colored process occurring on the left side of CPW1, the resulting spin waves propagate away from both CPWs making them undetectable. We will discuss later in detail the multiband four-magnon scattering model that results in two different processes while satisfying the momentum and energy conservation laws.

### Field evolution of coherent nonlinear spin waves

So far we discussed the signals obtained at the specific field of 28.6 mT. In Fig. 2, we show that the nonlinear spin wave spectra change significantly with field strength. We depict their power-dependent evolution within a narrow field range of 1.2 mT, which encompasses the field range for the nonadiabatic pumping regime as shown in Fig. 1c (see also Supplementary Fig. S4 for the measurement with extended field range). We summarize intensities obtained on both CPW1 (a) and CPW2 (b) while an RF signal with a frequency of $f_p$ = 4.25 GHz is applied to CPW1. The power-dependent spectra shown in Fig. 2a, b demonstrate that the high-frequency resolution of the VNA and the phase stability in frequency offset measurements allow us to detect unprecedentedly rich characteristics of spin waves in the nonlinear regime, which are coherent with the RF signal applied to CPW1. Such spin waves are a prerequisite for neuromorphic computing in magnonics. Here, we evidence their richness in an all-electrical measurement, allowing one to avoid the time-consuming readout (photon counting) in optical measurements and establish fully integrated nonlinear spin-wave networks.

We first separate the spectra into two groups. The area between the triangular dashed lines represents the low-field regime, which contains a magnon frequency comb (MFC)-like feature at low powers. Multiple discrete side peaks are evident at 28.0 mT. Also, between $P$ = 1 mW and about 3.5 mW, we observe a variable frequency spacing that reaches approximately 1.9 MHz at $P$ = 1 mW and 1.1 MHz at 2 mW (see Supplementary Fig. S6 for a detailed spectrum). Increasing power leads to enhanced intensity at the half frequency. This MFC-like behavior exhibits similarities to the multi-band scattering between adjacent magnon modes reported in recent studies[46–48]. The formation of multi-magnon bands is facilitated by the confinement of the magnetic system. In a study by Mohseni et al.[47], a frequency spacing of approximately 0.7 GHz was reported, which resulted from scattering between different laterally quantized modes in a 400 nm wide nanowire. In our work, we observe a much smaller spacing of a few MHz. This discrepancy in frequency spacing is attributed to the larger width of the YIG mesa used in our study ($w_{YIG}$ = 173 μm). Dividing by the geometric ratio of 173/0.4 ≈ 432, the frequency spacing is about 1.6 MHz, which is close to the measured frequency spacing in our experiment. While the MFC-like feature at low field is intriguing, its detailed frequency structure is not understood at the moment. In the following, we focus on the emerging nonlocal features at the high-field regime.

At 28.4 mT, the signal at $f_p/2$ abruptly vanishes with increasing power (note the drastic color change from red to white in log scale). Two side peaks emerge. The abrupt reduction of the $f_p/2$ peak at a (field-dependent) power value is a specific feature in the high-field regime. We discuss this in detail later in the "Discussion" section. This behavior is in contrast to ref. 25, where the generation of side peaks by four-magnon scattering was reported. However, the intensity at the main peak at $f_p/2$ remained strong. Note that the $f_p/2$ peak was still visible along with side peaks also in the low-field region of our experiment.

Surprisingly, the yellow dotted rectangle in Fig. 2b at 28.5 mT displays a different behavior. We observe the emergence of more branches at CPW2 (Fig. 2b) than at CPW1. These branches appear together with the previously discussed splitting of the half-frequency

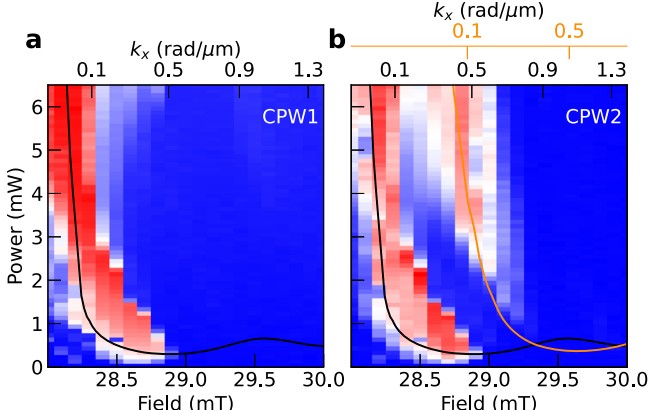

**Fig. 3 | Threshold characteristics of the nonlinear spin waves.** The intensities at $f_p/2$, as measured on **a** CPW1 and **b** CPW2, are plotted as a function of field and power. The top axis indicates the values $k_x$ relevant to excite $f_p/2$ calculated from the BVMSW mode. The minimum excitation power is reached at about 28.7 mT, which corresponds to $k_x = 0.5$ rad/µm (maximum CPW coupling efficiency). In (**b**), the signal from CPW2 is plotted. It contains an additional branch at a larger field and power. The black and orange solid lines show the calculated threshold power $P_{th}$ based on Eq. (1) with nonlinear magnon frequency shift of $\Delta f = 32$ MHz and $\Delta f = 0$, respectively. Two different top axes are shown for (**b**) as the differently colored curves are based on distinct magnon dispersions.

peak. The additional branches are closer in frequency to $f_p/2$ and merge at the frequency $f_p/2$ with increasing power. These excitations still induce weak signals at CPW1 visible at $f_p/2$ for high powers. At intermediate powers, while faint signals are visible on the high-frequency side of the CPW1 spectrum, these new branches are most prominently detected at CPW2 in Fig. 2b and have almost no counterpart in Fig. 2a from 28.5 to 29.0 mT at these power levels.

### Threshold characteristics

The threshold power for parametric excitation at $f_p/2$ is field-dependent and reaches a minimum value of about 0.2 mW at 28.7 mT as shown in Fig. 3a. At such a low VNA power, only the half-frequency peak appears. We note that the VNA port 2 connected to CPW2 can most efficiently detect induced voltages from magnons that are phase coherent with the pumping field (RF photon) applied at $f_p$ and possess $k_x = 0.5$ rad/µm. Indeed when calculating the value of $k_x$ to excite $f_p/2$ at each field (top axis in Fig. 3), the minimum power is reached for non-zero $k_x$ near 0.5 rad/µm. As shown in Fig. 1c, parametrically excited co-propagating spin waves exist in the regime of $\Delta H$ of approximately 1 mT consistent with the range of field for the branches displayed in Fig. 3a, b.

We now compare the measured data with a theoretical prediction. We evaluate $h_{th}$ derived in ref. 44:

$$h_{th} = \frac{v_g \arccos \tilde{h}_k}{w_{CPW} V_{kk} \sqrt{1 - \tilde{h}_k^2}}, \tag{1}$$

where $V_{kk}$ characterizes the parametric pumping efficiency[45], $w_{CPW}$ is the signal line width of 3.3 µm, and $\tilde{h}_k$ is the normalized antenna field profile as shown in Fig. 1d. At fixed $f_p$, the wave vector $k_p$ and the detected amplitude are expected to be field-dependent inside $\Delta H$ due the $k$-dependence of $h_k$. We converted the calculated values $h_{th}$ to power values $P_{th}$ using $P_{th} = \eta h_{th}^2$, where a scaling factor of $\eta = 4 \times 10^{-10}$ Ωm² is determined empirically to fit the experimental data accounting for uncertainties in the coupling efficiency and cable losses[49]. Calculated threshold curves as a function of field $H$ are plotted in Fig. 3. The black (orange) solid lines in Fig. 3 represent the calculated

values $P_{th}$ with (without) nonlinear magnon frequency shift, $\Delta f$. Introducing $\Delta f = +32$ MHz reproduces well the threshold behavior underneath CPW1 (black solid line) in Fig. 3a while the calculated $P_{th}$ with $\Delta f = 0$ (orange line) captures the additional branch observed in Fig. 3b. We note that the calculated curves $P_{th}(H)$ match the field-dependent intensities of the $f_p/2$ peaks well using only $\Delta f$ and $\eta$ as fitting parameters. The value of $\Delta f = +32$ MHz is a conservative estimate within the frequency shift range demonstrated in earlier works[10,50].

### Discussion

The characteristic spectra obtained by frequency-offset VNA spectroscopy are consistent with four-magnon scattering processes following nonadiabatic parametric pumping. Initially, magnons at half the pumping frequency, $f_p/2$, are pumped through the parametric pumping process. At higher power levels, the magnons at $f_p/2$ interact and scatter into adjacent magnon bands when satisfying the momentum and energy conservation laws. Four-magnon-scattering between co-propagating spin waves has been shown to occur after a characteristics delay time. The delay time is needed to compensate for the intrinsic damping of the secondary states[25,45,51]. Our data taken at fields 28.5 to 29.1 mT demonstrate that the four-magnon scattering does not take place at CPW1 where the highest excitation strength of spin-wave modes exists, but in-between CPWs when the parametrically pumped spin waves are on their way toward the detector. Strikingly the successively formed modes preserve phase coherency with the driving RF field and thus show a large signal-to-noise ratio at VNA port 2 when detected even 30 µm away from the emitter CPW. We note that the excitation by means of a non-synchronized external microwave generator with a stable phase led to magnon signals detected at port 2 of the VNA as well (see Supplementary Fig. S3). However, depending on the mode, the signal-to-noise ratio was significantly reduced and field-dependent spectra showed less fine structure.

To quantitatively model the field and power-dependent magnon frequency fine structures, numerical calculations were conducted to determine the solutions for the four-magnon scattering processes following the nonadiabatic parametric pumping. Two co-propagating magnons with their initial momenta and energies equivalent to $k_p/2$ and $f_p/2$ scatter with each other. Considering momentum and energy conservation for this four-magnon scattering process, it can be described as follows:

$$\begin{aligned} f_p/2 + f_p/2 &= f_1 + f_2, \\ k_{!p}/2 + k_{!p}/2 &= k_1 + k_2, \end{aligned} \tag{2}$$

where $f_1$ ($k_1$) and $f_2$ ($k_2$) represent the frequencies (wave vectors) of scattered magnons. The material parameters were extracted by performing further $S$-parameter measurements (see Supplementary Fig. S2). In our nonadiabatic pumping scheme, we directly pump modes with non-zero $k_p/2$ facilitating the 4-magnon scattering process. In contrast, achieving a comparable population of the non-zero $k_p/2$ mode in the homogeneous pumping regime would necessitate significantly higher power as the microwave pumping initially provides zero wave vector, leading to pairs of nonlinear magnons with opposite wave vectors. While the latter modes can undergo a subsequent 4-magnon scattering process, much higher power is needed because this process involves an additional step compared to our nonadiabatic pumping scheme.

Figure 4a illustrates the initial and final states at three different fields for the $n = 1$ intraband scattering process. Here, the parametrically pumped magnons (green filled dots) with frequency $f_p/2$ scatter into counter-propagating secondary magnons at $f_1$ and $f_2$ (blue open circles), all of which remain within the same band. The secondary magnons are detectable by both CPWs as they are counter-propagating. Interestingly, the intensity can vary depending on the frequency. On the one hand, the lower frequency magnon propagates

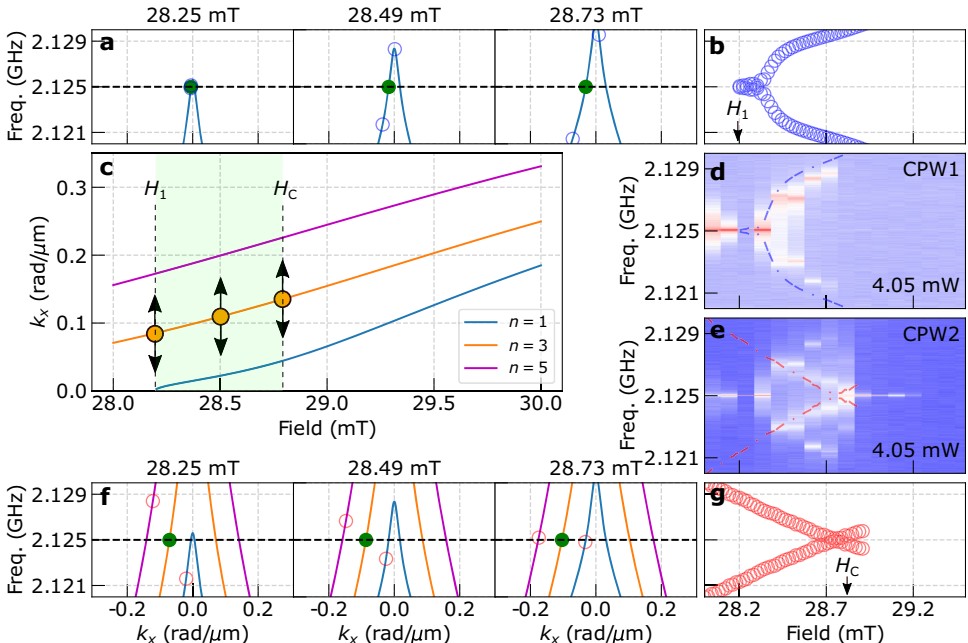

**Fig. 4 | Co-/counter-propagating magnon scattering model. a, f** Examples of initial and final states for scattering at three different fields. Green filled dots indicate the initial parametrically pumped magnons at **a** $n = 1$ band and **f** $n = 3$ band. Blue (red) open circles indicate the generation of counter (co)-propagating magnons after the four-magnon intra (inter)-band scattering in the panel (**a**) (**f**). Because the blue magnons propagate in opposite directions, they are detected in both CPWs. In contrast, red magnons co-propagate, therefore they are only detected in CPW2. This agrees with the experimental observation as shown in (**d, e**). Calculated field dependence of **b** the $n = 1$ intra-band and **g** the $n = 3 \rightarrow n = 1$ and $n = 5$ inter-band scattered magnon frequencies. These blue counter-propagating (red co-propagating) magnons show a field dependence, which diverges from (converges to) $f_p/2$. **c** Wave-vector-to-field relation for the BVMSWs

at $f_p/2$ with the discretization of wave vectors by the finite width $w_{Y\,IG}$ along the $y$-direction. Considering the symmetry of the antenna, we assume odd $n$ modes to be excited. $H_1$ is defined as a field above which the lowest $n = 1$ mode can exist. This $H_1$ is consistent with the lowest field found in (**b**). The interband magnon scattering is only possible above $H_1$. The scattering pairs are not found above $H_C$ as can be seen in (**g**). **d, e** Experimental data obtained on CPW1 and CPW2 as a function of field taken at 4.05 mW. Note the appearance of converging peaks only at CPW2. The observed field range of less than 1 mT is consistent with the expected field range for the interband scattering (green-shaded region in **c**). We use dashed blue and red lines indicating the calculated field dependence for the counter- and co-propagating processes, respectively to avoid reducing data visibility.

toward the right direction, which is toward CPW2 inducing a relatively strong intensity there. On the other hand, the upper-frequency one propagates toward the left resulting in a weaker signal at CPW2. This frequency-dependent signal distribution is complementary to the observation on CPW1 which provides a stronger (smaller) intensity for the upper (lower) frequency modes (see Fig. 1f). We also note that the half frequency magnon resides near $k_x = 0$, as shown in the left panel of Fig. 4a. Our CPW antenna coupling is zero at $k_x = 0$ suppressing the signal at half frequency. With a slight increase in the field, the dispersion shifts resulting in a non-zero $k_x$. This leads to an enhanced signal away from the half frequency. Figure 4b summarizes the field evolution of the side peak frequencies. The frequencies of these counter-propagating modes become farther separated from $f_p/2$ with increasing field as shown in Fig. 4b. In contrast, Fig. 4f depicts the interband scattering process from the parametrically pumped $n = 3$ to the $n = 1$ and $n = 5$ bands. Here the scattered magnons have the wave vectors of the same sign, indicating co-propagating behavior (red open circles). The frequency of these magnons comes closer to $f_p/2$ with increasing field as depicted in Fig. 4g.

Figure 4c shows $k_x$ and field values to excite $f_p/2$ for spin waves with $(n - 1)$ nodes along the $y$-direction. We considered only the three lowest $n = 1, 3$, and 5 modes considering the symmetry of our antenna and spin wave confinement imposed by the finite width $w_{YIG} = 173\,\mu m$ along the $y$-direction[52]. Note that thin YIG has shown a decay length of 860 μm in ref. 53. $H_1$ defines the field above which the $n = 1$ mode is occupied and the interband magnon scattering is allowed. $H_C$ denotes the field where no more scattering pairs are found (no solution exists). Only between $H_1$ and $H_C$, magnon scattering involving the $n = 1$ band is allowed.

The calculated characteristics are consistent with experiments displayed in Fig. 4d and e, where the calculated field dependencies (dashed lines) are superimposed on the measured CPW1 and CPW2 signals. We detect branches at both CPW1 and CPW2 which exhibit an increasing frequency separation from $f_p/2$ with increasing field. These branches are consistent with the counter-propagating magnons, marked in blue color in Fig. 4b. Interestingly, we see two branches merging with increasing field at $f_p/2$ only in the CPW2 data. This striking observation confirms that these coherently detected spin waves originate from the co-propagating magnons marked in red color in Fig. 4g. We refer the reader to Supplementary Fig. S1 for consistent trends observed at further power levels. We note that at 28.7 mT, our setup resolves clearly the remaining separation of only 2 MHz between the two branches merging in the CPW2 data in Fig. 4e. This demonstrates an unprecedented frequency resolution concerning propagating nonlinear spin waves far beyond BLS microscopy.

Finally, we discuss the evolution of nonlinear spectra depending on increasing and decreasing power levels to demonstrate data encoding via nonlinear processes. Here we used the same pumping frequency of $f_p = 4.25$ GHz as before. However, the nonlinear processes were observed at higher power levels above 6 mW and the magnetic resonance field was slightly lower. The modified parameters are most likely caused by a contact pad that was damaged and compromised the wave-impedance matching and RF current distribution. In Fig. 5, we show the spectra taken at a fixed field of 26.5 mT. We first swept the power up from 6 mW to 12 mW (Fig. 5a). Keeping the field constant, we then performed a second sweep in which we reduced the power from 12 mW to 6 mW (Fig. 5b). The RF field was continuously on during

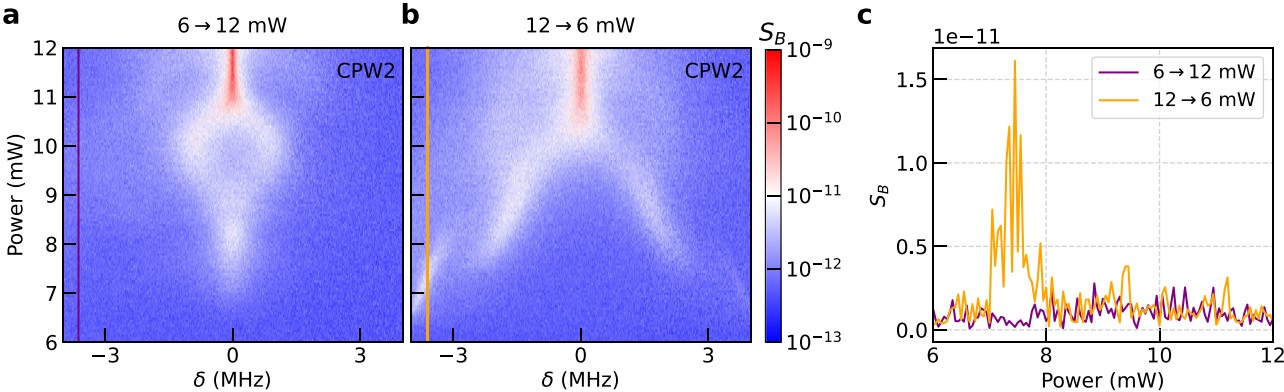

**Fig. 5 | Emergent nonlinear magnons dependent on power-sweep direction.** Transmission measurement across a YIG-based magnonic waveguide for sweeping the excitation power at $f_p$ = 4.25 GHz **a** from 6 mW to 12 mW and subsequently **b** from 12 mW to 6 mW. $\delta$ represents the deviation in detection frequency from $f_p$/ 2 = 2.125 GHz. **c** Power-history dependent signals extracted at $\delta$ = −3.6 MHz for the different sweep directions shown in (**a**) and (**b**). An emergent magnon mode appears at 7 mW only for the sweep starting at 12 mW down to 6 mW.

these measurements. Strikingly, we observe a distinct difference between the spectra of the first and second sweep. In Fig. 5c, we show two vertical line cuts at $\delta$ = −3.6 MHz, which demonstrate that an emerging magnon signal is pronounced at 7 mW with the second power sweep only (orange curve). A power-sweep dependent control of a nonlinear magnon signal was reported for inelastic light scattering experiments recently[10]. Using the same optical technique, Körber et al. demonstrated that different excitation pulse sequences resulted in different nonlinear magnon frequencies enabling a trainable magnonic system for neuromorphic computing[19]. Gartside et al. utilized a magnetic field-dependent memory effect to demonstrate the potential for efficient training using micromagnetic states[6]. In our experiments, we observe a power-dependent memory effect in parametrically pumped nonlinear magnons using an all-electrical technique with fast readout. This discussion illustrates the potential application of our nonlinear magnonic system for reservoir computing.

Our experiments and modeling evidenced that nonlinear spin wave modes were excited phase-coherently with the RF source. The nonlinear magnons were in phase coherency over at least 30 μm which was given by the distance between emitter and detector CPWs. The observed spatial coherency is consistent with a long temporal phase coherency reported in a very recent work[41] in which parametrically pumped nonlinear magnons were studied in a microstructured YIG film. Those data suggested that the magnons maintained phase coherency for a duration exceeding the relaxation time of linear modes by two orders of magnitude. We hence expect that the spatial coherency observed here is a lower limit and in fact considerably larger than the distance between the two integrated CPWs. Using all-electrical emission and detection, we identify 4-magnon scattering processes that are at the heart of the recently presented reservoir computing scheme with magnons[19]. The setup presented here significantly enhances the speed of signal readout and the frequency resolution compared to the established BLS microscopy. At the same time, the phase coherency of scattered magnons is accessible as a further data encoding parameter that goes beyond ref. [19]. Adding a pulsed and phase-synchronized external signal generator combined with a reference mixer, the phases and time delays of propagating nonlinear spin waves can be quantitatively determined all-electrically (see Supplemental Information for details about the phase measurement). Our findings direct the way to an integrated magnonic circuit for neuromorphic computing, which is interfaced with conventional and fast microwave electronics avoiding the time-consuming readout by inelastically scattered photons.

We have demonstrated the intricate spectral characteristics of parametrically excited coherent magnons using broadband microwave

spectroscopy with a frequency offset method. Our detection scheme exhibits remarkable sensitivity and frequency resolution, revealing the stringent frequency selection rule in parametric pumping and enabling the identification of extremely compact inter/intra-band scattering peaks separated by only about 2 MHz. The experimental observations closely align with the proposed theoretical model incorporating a four-magnon scattering process subsequent to a parametric pumping process. Notably, a new magnon branch emerges away from the source magnons, suggesting the nonlocal characteristics of propagating-magnon-driven 4-magnon scattering processes. The success of the presented study paves the way for the seamless integration of non-linear magnonics with high-frequency microwave electronics with highly dense coherent signals. This feature has received limited exploration thus far and holds great promise for future advancements in magnon-based computation and logic operations.

## Methods

### VNA frequency offset measurements

We utilized the frequency offset measurement option (opt 80) of our VNA (Keysight PNA N5222A). The RF source signal, operating at a frequency of $f_p$ = 4.25 GHz, was applied to VNA port 1. Both VNA ports 1 and 2 were connected to CPW1 and CPW2, respectively, via 50 Ω impedance coaxial cables and microwave probes. The front panel jumper of VNA port 2 was adjusted to direct 90% of the intensity to the mixer. The local oscillator was set to operate at an offset frequency $f_o$, swept between 2.12 GHz and 2.13 GHz in increments of 7 kHz. Mixed signals from the local oscillator and the nonlinear magnons were phase-sensitively detected by both Receivers A and B at CPW1 and CPW2, respectively. This setup enabled the detection of signals at frequencies $f_o$ deviating from the source frequency $f_p$.

### Device fabrication

Our sample consists of a single-crystalline, 130-nm-thick yttrium iron garnet (YIG) grown by liquid phase epitaxy on a 0.5-mm-thick gadolinium gallium garnet (GGG) substrate[42]. Two identical coplanar waveguides (CPWs), made from 180-nm-thick gold on a 5-nm-thick Ti adhesion layer, were fabricated using photolithography followed by the lift-off process. The CPWs have signal and ground lines, each 3.3 μm wide, with an edge-to-edge spacing of 2.7 μm. The center-to-center distance $d$ between the CPWs is 30 μm. The sample was shaped into a parallelogram using ion-beam etching, with the edges along the $y$-axis tilted by 10° (Fig. 1a), to prevent the formation of standing spin waves along the $x$-direction. The YIG device has lateral dimensions of $w_{YIG}$ = 173 μm and $\ell_{YIG}$ = 261 μm (Fig. 1a).

## Magnon dispersion analysis

We utilized the dispersion relations of Eq. (45) in ref. [52]. The following magnetic parameters were determined using standard $S$ parameter measurements on the YIG device: $\gamma/(2\pi) = 28$ GHz/T, $\mu_0 M_s = 0.176$ T, and $D_{ex} = 5.4 \times 10^{-17}$ T/m$^2$. We consider spin wave modes which have uniform profiles across the thickness, which ignores the surface pinning effect. This is indeed the case for our 130-nm-thick YIG film, where exchange interactions are dominant over dipolar effects. Higher-order perpendicular standing spin waves fall outside our detection frequency range.

Laterally, we implement a boundary condition that pins the spins at the edges running parallel to the $x$-axis, as dipolar pinning is anticipated to prevail over longer scales[54]. The symmetry in our coplanar waveguides (CPWs) ensures that only symmetric modes along the $y$-axis are detected, which forces the spin wave vector along $y$ to adopt values of $k_y = n\pi/w_{YIG}$ with $n$ being odd integers. When calculating the dispersion, we incorporated a systematic field calibration offset of $-0.75$ mT to account for the difference between actual and nominal magnetic field values. Then, we numerically search for possible scattering pairs that satisfy the momentum and energy conservation laws. We find that only a finite set of pairs can satisfy Eq. (2) due to the specific shape of the dispersion curves (Fig. 4a, f). We repeat this procedure with varying magnetic fields, which results in Fig. 4b, g.

## Data availability

The raw and calculated data for generating figures in the main text are provided in the Zenodo repository at https://doi.org/10.5281/zenodo.13134384.

## Code availability

The codes for generating spin wave dispersion, antenna field distribution, and searching for magnon-magnon scattering pairs are available on GitHub at https://github.com/kmogis/Emergent-NSW.

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

## Acknowledgements

We thank Kei Yamamoto, Grégoire de Loubens, and Kyung-Jin Lee for helpful discussions. We thank C. Dubs and P. Che for their support in sample fabrication. The research was funded by the EPFL COFUND Grant No. 665667 (K.A.), National Research Foundation of Korea, NRF-2021R1C1C2012269 (K.A.), and SNSF grant 197360 (D.G.).

## Author contributions

Designed the concept: D.G. Fabricated the samples: K.A. Performed the measurements: K.A., M.X. and A.M. Analyzed and interpreted the data: K.A., M.X., K.-W.M., C.H., and D.G. Performed the modeling: K.A., M.X., C.K., and K.-W.M. Prepared figures: K.A., M.X., and D.G. Wrote the main manuscript text: K.A., M.X., and D.G. All authors reviewed the manuscript and provided critical feedback.

## Competing interests

The authors declare no competing interests.
