## [Peer Review File · Nature Communications]

Emergent coherent modes in nonlinear magnonic waveguides detected at ultrahigh frequency resolutionREVIEWER COMMENTS

Reviewer #1 (Remarks to the Author):

In this manuscript, the authors reported the generation and detection of phase-coherent nonlinear magnons with complicated spectral characteristics in YIG. Combining the frequency-offset VNA spectroscopy measurements and magnon dispersion analysis, they demonstrated that the power- and field-dependent coherent nonlocal magnons exhibit fine structure, which can be well described by the inter-band four magnon scattering scheme they proposed. This work highlights the unique benefits of these nonlinear modes in future magnon-based computation and logic operations.

In summary, the work presented in this paper is interesting. Therefore, I recommend the acceptance of the paper, after fixing some critical points, listed as follows.

1. The authors claim that “the phase-coherency of scattered magnons is accessible as a further data encoding parameter” and “Our findings direct the way to an integrated magnonic circuit for neuromorphic computing”. Can the authors perform the experimental measurements or propose detailed strategies to realize the function of data encoding or neuromorphic computing?
2. The authors conduct the investigations in the nonadiabatic condition. When the pumping area is larger than the wavelength of the spin wave, are the results in the manuscript still the same? Please discuss the distance that the nonlinear magnons can keep the phase coherence.
3. Please explain why the signal at $fp/2$ abruptly vanishes with increasing power with two side peaks emerging, in contrast to the case that the intensity at the main peak at $fp/2$ remained strong.

Reviewer #2 (Remarks to the Author):

This study investigates an Yttrium Iron Garnet (YIG) film coated on a gadolinium gallium garnet (GGG) substrate, where two coplanar waveguides (CPWs) are positioned on the YIG film to facilitate both the excitation and detection of magnons. Utilizing a Vector Network Analyzer (VNA) in frequency offset mode, the authors assert the detection of nonlinear magnons, phase-sensitively, propagating a macroscopic distance (30 μ m) between the CPWs.

The research examines both counter-propagating and co-propagating magnon scattering processes. This seems akin to measuring reflection and transmission coefficients. By maintaining a constant magnetic field of 28.6 mT and varying the power of the microwave excitation in port 1, the authors observed a distinct peak at powers below 1.7 mW. Above this power threshold, the spectrum exhibited a pair of side peaks. Notably, an additional set of side peaks was detected exclusively by Receiver B in port 2 at powers exceeding 1.7mW. The researchers “attribute” the side peaks detected at both ports to counter-propagating spin waves, while those observed only at port 2 to co-propagating spin waves. Furthermore, they also explored the field evolution of these spin waves, revealing intriguing results such as the MFC-like feature in the low-field regime.

Despite the experimental techniques, the paper exhibits several significant shortcomings. First, the paper falls short of demonstrating the level of novelty expected for a highly selective journal like Nature Communications. Even though magnon propagation might captivate specialists in magnetism, the overall innovation seems lacking. Second, from an experimental perspective, the results are not convincing. A significant issue is the absence of thorough explanation and rigorous analytical support for the data, the findings are not elucidated with the necessary rigor and determinism expected in an experimental article, which is a fundamental requirement for credible scientific report. For example, the claim that side peaks in Figures 1f and 1g result from a 4-magnon scattering process is not substantiated by any quantitative analysis or data fitting, which is a critical omission in their manuscript. One extra example, some of the key explanations rely solely on qualitative references to external sources (e.g., Ref [42] in lines 110, 119, 129, 214), undermining the strength of their conclusions. The frequent use of vague terms such as “roughly” (line 169) and “reminiscent” (line 184) is atypical in experimental research papers and indicates a general lack of precision in explaining the experimental data. Moreover, the repetitive use of subjective phrases like “we attribute” or “be attributed to” (appearing seven times in the main text) to explaining further exemplifies the ambiguous interpretation of the experimental findings. These issues collectively contribute to the paper's inability to meet the rigorous standards of scientific discourse expected in a scientific journal like Nature Communications.

However, even if the experimental data are elucidated in a more compelling manner, I believe that the results presented in this paper still fall short of the stringent criteria required for the publication in Nature Communications. The paper would be better suited for a more specialized focus journal.

Consequently, I do not recommend it for publication in Nature Communications.

Reviewer #3 (Remarks to the Author):

This work proposes a nontrivial method to detect the nonlinear spin-wave signals in YIG thin films by utilizing the frequency offset measurement option of VNA. The frequency resolution can approach to kHz range in the experiment, which is unprecedented in my point of view. The local and nonlocal behavior of the nonlinear signals is systematically discussed, especially the four-magnon scattering process. This work offers an advanced pathway for studying the nonlinear behaviors of magnons, and the high frequency resolution enables further investigation of the magnon frequency combs in various materials. The data are convincing and the model is reasonable and consistent with experiments. Thus, I recommended the publication of this work in Nature Communications with minor revisions. Following are my comments and suggestions:

- 1) The title mentioned "magnon circuits" which is quite confusing because there are no circuit designs or prototype devices in the main text. I recommend the author to reconsider the title.
- 2) The authors mentioned that the signal they measured is phase-sensitive voltages. However, I cannot get any phase information from the VNA data. Could the authors comment on this?
- 3) In the discussion, the author mentioned the three-magnon process is the mechanism that induces the frequency splitting from f_p to $f_p/2$. However, from the dispersion shown in Fig.

1c, there is no magnon band at ω_p . Since the three-magnon process should involve "three magnons", the process described in this work is attributed to the nonlinear scattering between the microwave photons and magnons, namely, the "parametric pumping". Please confirm this point in the main text.

4) The authors detected two kinds of four-magnon scattering modes in the reflection and transmission spectra and I think the results are very interesting and novel. From the scattering process diagram shown in Fig. 4c, the counter-propagating process requires higher Δk than the co-propagating process for momentum conservation, which means it needs larger critical power. However, from the power-dependent measurements shown in Fig. 2 and Fig. 3, the co-propagating process needs a higher power to trigger, which is contradictory.

5) The CPW used in this work is in the size of micrometers, thus the excitation k value is not large. I wonder why the authors chose such size rather than nanometer-scale CPW, which has been usually used in the author's previous work. It could be more interesting to use a nanometer-size CPW to excite higher k value magnons, which could cover the valley position of MSBVM dispersions.

6) The Fig. 1a is compressed, so please adjust it.

Some typos:

7) Line 44, "synchronization" should be "synchronization"

8) Caption in Fig 4, "co-propogating" should be "co-propagating"

9) Line 263, "quantatively" should be "quantitatively"

Responses to Reviewers

The reviewers' comments are indicated in blue color, our answers are in black. Changes are highlighted in green in the main text and supplementary materials.

1 Reviewer 1

In this manuscript, the authors reported the generation and detection of phase-coherent nonlinear magnons with complicated spectral characteristics in YIG. Combining the frequency-offset VNA spectroscopy measurements and magnon dispersion analysis, they demonstrated that the power- and field-dependent coherent nonlocal magnons exhibit fine structure, which can be well described by the inter-band four magnon scattering scheme they proposed. This work highlights the unique benefits of these nonlinear modes in future magnon-based computation and logic operations. In summary, the work presented in this paper is interesting. Therefore, I recommend the acceptance of the paper, after fixing some critical points, listed as follows.

Answer: We thank the reviewer for the careful reading and the positive evaluation of our manuscript. We thank for recommending acceptance after appropriate revision.

1. The authors claim that “the phase-coherency of scattered magnons is accessible as a further data encoding parameter” and “Our findings direct the way to an integrated magnonic circuit for neuromorphic computing”. Can the authors perform the experimental measurements or propose detailed strategies to realize the function of data encoding or neuromorphic computing?

Answer: To substantiate that data encoding via nonlinear magnon excitation is possible in our setup, we investigated nonlinear spectra depending on increasing and decreasing power levels. In Fig. R1a we show spectra at a fixed field of 26.5 mT when we first swept the power from 6 mW to 12 mW. Secondly, we performed a subsequent power sweep from 12 mW to 6 mW shown in Fig. R1b. The rf field was continuously applied to CPW1 during the experimental measurements. Importantly, we observed a distinct difference between the first and second sweep. In Fig. R1c, we show two vertical linecuts at $\delta = -3.6$ MHz. The direct comparison demonstrates that a nonlinear magnon branch emerges in the 2nd sweep (orange curve) near 7.5 mW which is clearly absent in the first one (magenta curve). Considering Refs. [Q. Wang et al., *Sci. Adv.*, 9, eadg4609 (2023)] and [L. Körber et al., *Nature Communications*, 14(1), 3954 (2023)] such power-history dependent behavior is a key prerequisite for enabling trainable magnonic systems for neuromorphic computing with nonlinear spin waves. In contrast to these earlier works, we observe the power-history dependent memory effect in an all-electrical measurement setup, which avoids the additional readout based on time-consuming photon counting in inelastic light scattering microscopy. We added Fig. R1 as Fig. 5 to the main text. We thank the reviewer for the suggestion.

2-1. The authors conduct the investigations in the nonadiabatic condition. When the pumping area is larger than the wavelength of the spin wave, are the results in the manuscript still the same?

Answer: To realize an experimental scenario with our existing CPW on the YIG film and answer the question given by the reviewer one would need to apply a much higher magnetic field such that the dashed line in Fig. 1c intersects the BVMSW dispersion relations at much higher k_x values. Figure 3a and 3b show that our sensitivity is mostly limited to the spin waves with k_x values near the first minimum as seen in the Fourier analysis in Fig. 1d. The wavelengths much smaller than the pumping area as addressed by the reviewer's question are not detectable with our current CPW. The investigation of ultra-short wavelengths could be the subject of a separate study going beyond the scope of the present manuscript.

Figure R1: **Emergent nonlinear magnon signals dependent on power-sweep direction** Transmission measurement across a YIG-based magnonic waveguide for sweeping the excitation power at $f_p = 4.25$ GHz **a** from 6 mW to 12 mW and, subsequently, **b** from 12 mW to 6 mW. δ represents the deviation in detection frequency from $f_p/2 = 2.125$ GHz. **c** Power-history dependent signals extracted at $\delta = -3.6$ MHz for the different sweep directions shown in (a) and (b). An emergent magnon mode appears at 7.5 mW only for the sweep starting at 12 mW down to 6 mW. The VNA power levels are higher than in the main text most likely caused by a severely scratched pad which compromised the wave impedance matching.

While our existing setup does not allow to investigate the question raised by the reviewer, we speculate about the behavior expected for a larger pumping area. The emergence of the fine structure in frequency, which we attribute to 4-magnon scattering processes following parametric pumping, requires a population of $f_p/2$ magnons with non-zero wave vector $k_p/2$. For a large pumping area, we expect that a significantly higher power would be needed to achieve a comparable population of the nonzero $k_p/2$ mode, as the microwave pumping would initially provide zero wave vector and scattering into pairs with opposite wave vectors occurs. These nonzero k modes are counter-propagating and need to undergo further 4-magnon scattering to provide a large population at $k_p/2$ propagating towards CPW2. This process would require much higher power because it contains an additional step compared to the nonadiabatic condition explored in our work, in which we directly pump modes with non-zero wave vectors $k_p/2$ facilitating the detection of the 4-magnon scattering process. We added the discussion to the revised version of the manuscript.

2-2. Please discuss the distance that the nonlinear magnons can keep the phase coherence.

Answer: Experimentally we observe that nonlinear magnons are in phase coherency over $30 \mu\text{m}$ which is the distance between emitter and detector CPW. This observation suggests that the distance mentioned by the reviewer is longer than $30 \mu\text{m}$. In a first approximation, the magnon decay length can be estimated from the Gilbert damping parameter, where the magnon relaxation time is given by $\tau_r = 1/(\alpha\omega)$. Using $\alpha = 2 \times 10^{-4}$ for our YIG film, we obtain $\tau_r = 375 \text{ ns}$ at $\omega = 2\pi \times 2.125 \text{ GHz}$. When this is multiplied with the magnon group velocity of 350 m/s , we obtain $130 \mu\text{m}$ for the decay length of a backward volume magnetostatic wave after the scattering process. However, considering a very recent publication, we assume this estimation to provide a minimum distance value. T. Makiuchi et al. [Nature Materials 23, 627 (2024)] demonstrated the recall of the magnetization-precession phase after times that exceeded the relaxation timescale by two orders of magnitude when studying parametric pumping of nonlinear magnons in a microstructured YIG film. We added the discussion and reference to the revised version of the manuscript.

3. Please explain why the signal at $f_p/2$ abruptly vanishes with increasing power with two side peaks emerging, in contrast to the case that the intensity at the main peak at $f_p/2$ remained strong.

Answer: In the original version of our manuscript, we had stated: “The abrupt reduction of the $f_p/2$ peak at a (field-dependent) power value is a specific feature in the high-field regime. We attribute the transition to a threshold process for which the excitation of $f_p/2$ has a higher threshold than the excitation of the side peaks”. Here we provide more detailed explanation. The diverging branch with field can be explained by the intra-band counter-propagating magnon scattering, where all the magnon pairs are located in the $n = 1$ band (see Fig. R2a). From the left dispersion curves, one sees that the half-frequency magnon resides at almost $k_x = 0$, where our CPW antenna coupling efficiency is zero (see Fig. 1d in the main text). This leads to a vanishing intensity at half frequency. With a slight increase of the field the dispersion shifts leading to non-zero k_x , which has better coupling efficiency to our CPW antenna. Therefore the signal intensity increases with increasing field, but their frequencies move away from the half frequency (panel b). The panel d of Fig. R2 compares the expected signals (blue dashed lines) with the measured data. This scenario successfully explains the vanishing half frequency peak at the splitting point and the emerging signals. We modified Fig. 4 and added discussion in the manuscript. We thank the reviewer for raising this important point.

2 Reviewer 2

This study investigates an Yttrium Iron Garnet (YIG) film coated on a gadolinium gallium garnet (GGG) substrate, where two coplanar waveguides (CPWs) are positioned on the YIG film to facilitate both the excitation and detection of magnons. Utilizing a Vector Network Analyzer (VNA) in frequency offset mode, the authors assert the detection of nonlinear magnons, phase-sensitively, propagating a macroscopic distance (30 μ m) between the CPWs.

The research examines both counter-propagating and co-propagating magnon scattering processes. This seems akin to measuring reflection and transmission coefficients. By maintaining a constant magnetic field of 28.6 mT and varying the power of the microwave excitation in port 1, the authors observed a distinct peak at powers below 1.7 mW. Above this power threshold, the spectrum exhibited a pair of side peaks. Notably, an additional set of side peaks was detected exclusively by Receiver B in port 2 at powers exceeding 1.7mW. The researchers “attribute” the side peaks detected at both ports to counter-propagating spin waves, while those observed only at port 2 to co-propagating spin waves. Furthermore, they also explored the field evolution of these spin waves, revealing intriguing results such as the MFC-like feature in the low-field regime.

Despite the experimental techniques, the paper exhibits several significant shortcomings. First, the paper falls short of demonstrating the level of novelty expected for a highly selective journal like Nature Communications. Even though magnon propagation might captivate specialists in magnetism, the overall innovation seems lacking.

Answer: We thank the reviewer for carefully reading our work. However, we respectfully disagree with the reviewer’s assessment regarding the lack of innovation. First of all, we emphasize that the nonlinear magnonics are of great fundamental importance in constructing energy-efficient neuromorphic computation systems due to their inherently low nonlinear operation energy consumption, down to the attojoule (aJ) level [X. Liang et al. Nature Electronics (2024): 1-14]. This has driven tremendous research efforts and attracted significant attention as evidenced by recent works published in journals with broad readership:

- T. Makiuchi et al. "Persistent magnetic coherence in magnets." Nature Materials 23 (2024): 627–632.
- O. Lee et al. "Task-adaptive physical reservoir computing." Nature Materials 23 (2024): 79-87.
- C. Koerner et al. "Frequency multiplication by collective nanoscale spin-wave dynamics." Science 375

Figure R2: **Co-/Counter-propagating magnon scattering model.** **a,f** Examples of initial and final states for scattering at three different fields. Green filled dots indicate the initial parametrically pumped magnons at **a** $n = 1$ band and **f** $n = 3$ band. Blue (red) open circles indicate the generation of counter (co)-propagating magnons after the four-magnon intra (inter)-band scattering in the panel **a** (**f**). Because the blue magnons propagate in opposite directions, they are detected in both CPWs. In contrast, red magnons co-propagate, therefore they are only detected in CPW2. This agrees with the experimental observation as shown in **d,e**. Calculated field dependence of **b** the $n = 1$ intra-band and **g** $n = 3 \rightarrow n = 1$ & $n = 5$ inter-band scattered magnon frequencies. These blue counter-propagating (red co-propagating) magnons show a field dependence, which diverges from (converges to) $f_p/2$. **c** Wave-vector-to-field relation for the BVMSWs at $f_p/2$ with the discretization of wave vectors by the finite width w_{YIG} along the y -direction. Considering the symmetry of the antenna, we assume odd n modes to be excited. H_1 is defined as a field above which the lowest $n = 1$ mode can exist. This H_1 is consistent with the lowest field found in **b**. The interband magnon scattering is only possible above H_1 . The scattering pairs are not found above H_C as can be seen in **g**. **d,e** Experimental data obtained on CPW1 and CPW2 as a function of field taken at 4.05 mW. Note the appearance of converging peaks only at CPW2. The observed field range of less than 1 mT is consistent with the expected field range for the interband scattering (green-shaded region in **c**). We use dashed blue and red lines indicating the calculated field dependence for the counter- and co-propagating processes, respectively to avoid reducing data visibility.

(2022): 1165-1169.

- L. Körber et al. "Pattern recognition in reciprocal space with a magnon-scattering reservoir." *Nature Communications* 14 (2023): 3954.

- Q. Wang et al. "A magnonic directional coupler for integrated magnonic half-adders." *Nature Electronics* 3.12 (2020): 765-774.

Each of these works addresses a different aspect of nonlinear spin-wave excitation, but none of them reports the discovery of the spatial coherence of propagating magnons after 4-magnon scattering events. Using an all-electrical readout of propagating spin waves our results have direct implications for the development of practical magnonic computing by avoiding the time-consuming photon-counting applied earlier and by allowing for an open system and thereby cascaded neural networks (see below). After submitting our work on the first observation of long-distance coherency of nonlinear magnons in thin-film YIG the high-impact work of T. Makiuchi et al. [*Nature Materials* 23, 627 (2024)] was published. While these authors investigated temporal coherency in a single isolated YIG disk, we investigated the spatial coherency along the propagation path of nonlinear magnons enabling the processing of nonlinear signals, which is a critical ingredient to enable connectivity in a full magnonic circuit. We implemented changes in the main text to further emphasize the novelty of our work.

Second, from an experimental perspective, the results are not convincing. A significant issue is the absence of thorough explanation and rigorous analytical support for the data, the findings are not elucidated with the necessary rigor and determinism expected in an experimental article, which is a fundamental requirement for credible scientific report. For example, the claim that side peaks in Figures 1f and 1g result from a 4-magnon scattering process is not substantiated by any quantitative analysis or data fitting, which is a critical omission in their manuscript. One extra example, some of the key explanations rely solely on qualitative references to external sources (e.g., Ref [42] in lines 110, 119, 129, 214), undermining the strength of their conclusions.

Answer: Stimulated by the reviewer's feedback, we extended our quantitative analysis. In the original version of our manuscript, we already presented a good quantitative agreement in some parts. For example, we had reported a fully quantitative agreement between measurement and analytic calculation in Fig. 3. We go beyond this in the revised manuscript.

In Fig. R2 (which is our new Fig. 4 in the main text) we present secondary magnon pairs which consider the following momentum and energy conservation process: $2k_0 = k_1 + k_2$ and $f_p = f_1 + f_2$, where k_0 is the wave vector corresponding to the intersection of the dispersion curve with the $(f_p/2)$ frequency. k_1 and k_2 represent the magnon wave vectors at secondary magnon bands. The green filled dots in Fig. R2(a,f) represent the parametrically excited magnon pairs at $f_p/2$ with wavevectors of k_0 . For the subsequent events, we have identified two different magnon scattering processes, i.e., intraband counter-propagating (Fig. R2a and b) vs. interband co-propagating magnons (Fig. R2f and g). In Fig. R2a, one can see that the initial magnon (green dots) and two secondary magnons (blue open circles) are all within a single band ($n = 1$) and counter-propagating (except for the negligibly small wavevector at 28.25 mT). In Fig. R2f, however, while the initial magnons are at $n = 3$ band (green dots), the secondary magnons (red open circles) are at different bands ($n = 1$ and $n = 5$). Here the scattering is an inter-band process and two secondary magnons are co-propagating. The field evolution of these two processes (panels b and g) are well matched with experimentally observed behavior as seen in Fig. R2d and e.

Figure R2c provides additional insight in fields and wavevectors giving rise to these scattering processes, where three lines represent parameters for which the magnon frequency becomes $f_p/2 = 2.125$ GHz with three different n values. One can see that the $n = 1$ is populated only above $\mu_0 H_1 = 28.2$ mT. This H_1 value

determines the emergence of side peak structure. The side peak structure disappears at $\mu_0 H_C = 28.8$ mT, above which no scattering pairs are found anymore (see Fig. R2g). This agrees well with the field range observed in the experiments (Fig. R2d and g).

To explain the field evolution of spectra at different powers, we further considered the nonlinear magnon frequency shift, which was introduced in earlier works [F. Guo et al., Phys. Rev. B 91 064426 (2015) and A. Slavin et al., IEEE Trans. Mag. 45 1875 (2009)]. We use the following empirical formula: $\Delta f = P[a(\mu_0 H)^2 + b(\mu_0 H) + c]$, where $a = -2.8114 \times 10^{-4}$ GHz/(mT²·mW), $b = 8.1363 \times 10^{-3}$ GHz/(mT·mW), and $c = 5 \times 10^{-4}$ GHz/mW were determined to reproduce the measured field dependence at different powers. In Fig. R3, we show the field dependent spectra at five different powers obtained for CPW1 and CPW2. We compare the measured spectral evolution with our calculation based on the power dependence. Several of the observed branches show a good quantitative agreement.

Figure R3: **Field dependent spectra with various excitation powers.** **a** CPW1 and **b** CPW2 signals detected by the VNA for nonlinear spin waves at different power levels as a function of in-plane field applied perpendicular to the CPWs. The excitation frequency was $f_p = 4.25$ GHz. The nonlocal magnon branch starts to emerge above 1 mW as seen in the CPW2 spectra. The field for splitting shifts down with increasing power, consistent with the shift shown in Fig. 2 of the main text. Correspondingly the second converging point in the CPW2 spectra also shifts down with power. Dashed blue and red lines are the calculated field dependence for the counter- and co-propagating processes, respectively

We further improved our Fig. 3. Stimulated by earlier works [Q. Wang et al., Sci. Adv. 9 eadg4609 (2023), D. Breitbach et al., Appl. Phys. Lett. 124 092405 (2024), and Q. Wang et al., arXiv:2403.13276 (2024)] we assume that the strong rf field drives the nonadiabatic parametric pumping process directly underneath CPW1 and leads locally to a frequency shift, Δf for the magnon dispersion relation. Away from CPW1, the magnon frequency shift disappears. This earlier presented scenario produces an inhomogeneous spatial profile of magnon frequencies. The black (orange) solid line in Fig. R4 represents the calculated threshold power values P_{th} with (without) magnon frequency shift, Δf . Introducing $\Delta f = +32$ MHz reproduces well the threshold behavior underneath CPW1 (black solid line) while P_{th} with $\Delta f = 0$ reproduces the orange line in Fig. R4b reflecting CPW2. The calculated P_{th} 's match well the field-dependent intensity of the $f_p/2$ peak with Δf as a fitting parameter. The value of $\Delta f = +32$ MHz is reasonable and smaller than demonstrated in the earlier research. We note that coming from small fields the measured signal disappears near the first local minimum in the calculated P_{th} . This is because no secondary magnon pairs are found above H_C as can be seen in Fig. R2g.

Figure R4: **Threshold characteristics of the nonlinear spin waves.** The intensities at $f_p/2$ as measured on **a** CPW1 and **b** CPW2 are plotted as a function of field and power. The top axis indicates the values k_x relevant to excite $f_p/2$ calculated from the BVMSW mode. The minimum excitation power is reached at about 28.7 mT, which corresponds to $k_x = 0.5 \text{ rad}/\mu\text{m}$ (maximum CPW coupling efficiency). In **b**, the signal from CPW2 is plotted. It contains an additional branch at larger field and power. The black and orange solid lines show the calculated threshold power P_{th} based on Eq. 1 with nonlinear magnon frequency shift of $\Delta f = 32 \text{ MHz}$ and $\Delta f = 0$, respectively. Two different top axes are shown for **b** as the two solid lines are based on distinct magnon dispersions

Considering the extended quantitative comparison we have revised Figs. 3 and 4 and the discussion about different scattering processes.

The frequent use of vague terms such as “roughly” (line 169) and “reminiscent” (line 184) is atypical in experimental research papers and indicates a general lack of precision in explaining the experimental data. Moreover, the repetitive use of subjective phrases like “we attribute” or “be attributed to” (appearing seven times in the main text) to explaining further exemplifies the ambiguous interpretation of the experimental findings. These issues collectively contribute to the paper’s inability to meet the rigorous standards of scientific discourse expected in a scientific journal like Nature Communications.

Answer: Stimulated by the reviewer we first reviewed the definitions of “attribute to” using (a) <https://www.dictionary.com/browse/attribute>: “attribute”: to regard as resulting from a specified cause; consider as caused by something indicated (usually followed by to): example: She attributed his bad temper to ill health; and (b) British dictionary: attribute to - to regard as belonging (to), produced (by), or resulting (from); ascribe (to): example to attribute a painting to Picasso. Second, we searched for “attribute to” on the Nature Communications homepage where one can browse through articles. Here, we got 800 results for our search. This high number suggests that the term “attribute to” is well accepted in the journal as it specifies a cause (see above) for an observation etc.. We have also checked our manuscript and then found that we used “roughly” and “reminiscent” only once for each. Following the reviewer’s suggestion, we still rephrased the sentences.

However, even if the experimental data are elucidated in a more compelling manner, I believe that the results presented in this paper still fall short of the stringent criteria required for the publication in Nature Communications. The paper would be better suited for a more specialized focus journal.

Consequently, I do not recommend it for publication in Nature Communications.

Answer: Our results represent a significant advancement in that we demonstrate the spatial coherence of nonlinear magnons promising a practical application. In typical BLS experiments [e.g. see M. Mohseni et al., *physica status solidi (RRL)*–Rapid Research Letters 14 2000011 (2020)], only the half frequency peak at $f_p/2$ is observed. However, with a single frequency component, magnon neuromorphic computing is impossible. Multiple frequencies were found in disk structures [L. Körber et al. *Nature Communications* 14, 3954 (2023)]. Although a disk could be used as a magnonic memory, magnonic information transfer is not possible within the confined magnetic structure. Our experiment is based on a spin-wave waveguide allowing for the transport of information via propagating magnons encoded in multiple different frequency components. In the revised manuscript, we further demonstrate the power-dependent memory effect in parametrically pumped propagating magnons. Our semi-open waveguide system is a new avenue and enables a cascaded neuromorphic computing platform with fast electronic readout. The rigorous analysis, which we performed as stimulated by the reviewer 2, demonstrates that our model captures the quantitative aspect of the experimental results. We are now confident that this comprehensive analysis combined with potential impact of our work in magnonic computing does meet the high standards of Nature Communications.

3 Reviewer 3

This work proposes a nontrivial method to detect the nonlinear spin-wave signals in YIG thin films by utilizing the frequency offset measurement option of VNA. The frequency resolution can approach to kHz range in the experiment, which is unprecedented in my point of view. The local and nonlocal behavior of the nonlinear signals is systematically discussed, especially the four-magnon scattering process. This work offers an advanced pathway for studying the nonlinear behaviors of magnons, and the high frequency resolution

enables further investigation of the magnon frequency combs in various materials. The data are convincing and the model is reasonable and consistent with experiments. Thus, I recommended the publication of this work in Nature Communications with minor revisions. Following are my comments and suggestions: 1) The title mentioned "magnon circuits" which is quite confusing because there are no circuit designs or prototype devices in the main text. I recommend the author to reconsider the title.

Answer: We thank the reviewer for the positive evaluation and constructive feedback. When creating the title we were inspired by a recent roadmap [A. Chumak et al., IEEE Transactions on Magnetics 58 1 (2022)] which stated that "spin-wave conduits are the fundamental elements for guiding spin waves and the realization of magnonic circuits.". However, considering the comment by the reviewer, we have now changed our title. It reads: "Emergent coherent modes in nonlinear magnonic waveguides detected at ultrahigh frequency resolution".

2) The authors mentioned that the signal they measured is phase-sensitive voltages. However, I cannot get any phase information from the VNA data. Could the authors comment on this?

Answer: In the current setup the measured signals depends on phase coherency but we do not measure the phases of nonlinear spin waves quantitatively. Measuring these phases of the scattered magnons would certainly enable further data encoding (beyond the usage of the amplitudes demonstrated above). In the revised manuscript we now comment on the strategy to access the phase information of the coherent parametrically pumped magnons. The extended setup requires an additional reference mixer as follows:

- (1) Apply an input RF signal at 4.25 GHz and split it into the YIG path (integrated CPW) and a reference-mixer path.
- (2) Tune the reference-mixer frequency to match the frequency of interest from the YIG output (e.g., 2.125 GHz).
- (3) Combine the reference signal and the magnon signal to determine their relative phase.

We included the phase acquisition method in the supplemental documents to help readers who are interested in acquiring phase information of nonlinear magnons.

3) In the discussion, the author mentioned the three-magnon process is the mechanism that induces the frequency splitting from f_p to $f_p/2$. However, from the dispersion shown in Fig. 1c, there is no magnon band at f_p . Since the three-magnon process should involve "three magnons", the process described in this work is attributed to the nonlinear scattering between the microwave photons and magnons, namely, the "parametric pumping". Please confirm this point in the main text.

Answer: The reviewer is correct that there is no magnon branch as an eigenstate of the system. In Fig. 1 of the original manuscript we used indeed the term "parametric pumping" suggested by the reviewer. Specifically we described the process as "nonadiabatic parametric pumping via forced spin precession at f_p and k_p ". In the remaining manuscript we then considered this type of "forced precession" as a "virtual" magnon following the concept introduced by a recent work [Victor S. L'vov et al., Phys. Rev. Lett. 131, 156705 (2023)]. We now see that this aspect generated confusion. In the revised version we have followed the reviewer's suggestion and changed "three-magnon process" to "parametric pumping process".

4) The authors detected two kinds of four-magnon scattering modes in the reflection and transmission spectra and I think the results are very interesting and novel. From the scattering process diagram shown in Fig. 4c, the counter-propagating process requires higher Δk than the co-propagating process for momentum conservation, which means it needs larger critical power. However, from the power-dependent measurements

shown in Fig. 2 and Fig. 3, the co-propagating process needs a higher power to trigger, which is contradictory.

Answer: The reviewer is asking about the required power to excite counter-propagating and co-propagating magnons. Our improved analysis in quantitative terms (see above) reveals that the diverging branch with respect to field actually originates from the intra-band magnon scattering within only the $n = 1$ band. This process contains smaller counter-propagating k vectors (Fig. R2a) compared to the interband-copropagating magnon scattering process (Fig. R2f). Indeed the required power to excite co-propagating magnons is higher as the reviewer noted. This can be seen from the spectrum at 28.9 mT in Fig. 2. The new branch in CPW2 (co-propagating magnons) appears at higher power than the splitting point (counter-propagating magnons). Our improved analysis is consistent with the reviewer's comment.

5) The CPW used in this work is in the size of micrometers, thus the excitation k value is not large. I wonder why the authors chose such size rather than nanometer-scale CPW, which has been usually used in the author's previous work. It could be more interesting to use a nanometer-size CPW to excite higher k value magnons, which could cover the valley position of MSBVM dispersions.

Answer: In this work, we focus on the generation and exploration of nonlinear magnons propagating over macroscopic distances in a spin-wave waveguide used in magnonic circuits. The width and correspondingly transferred wave vectors were such that relatively large group velocities of a few 100 m/s were realized. Correspondingly, we expected a large decay length. In the minimum of the MSBVM dispersion relation, the group velocity is (near) zero and the decay length of magnons is ultrashort compared to the distance between CPWs. This is the opposite limit which makes the detection of rich spectra at CPW2 unlikely due to the severe decay.

6) The Fig. 1a is compressed, so please adjust it.

Answer: We adjusted Fig. 1a. We thank the reviewer for pointing this out.

Some typos:

7) Line 44, "synchronization" should be "synchronization"

8) Caption in Fig 4, "co-prorogating" should be "co-propagating"

9) Line 263, "quantatively" should be "quantitatively"

Answer: Thank you for point out the typos. We corrected them.

REVIEWER COMMENTS

Reviewer #1 (Remarks to the Author):

The authors well addressed the points raised by the reviewers. I now recommend the publication of this paper in Nature Communications.

Reviewer #2 (Remarks to the Author):

The authors have now convinced me that the innovation presented in this work is sufficient for publication in Nature Communications, with the addition of quantitative analysis. I can now recommend it for publication, provided that the following questions are addressed:

1. In Fig. 1f and 1g, why are the linewidths small when the microwave drive power is low and significantly larger when the microwave drive power is high? How is this related to the Gilbert damping of the material and the drive power?
2. I noticed a signal (peak) at the center in Fig. 1f (also in Fig. 2a at 28.6 mT) when the drive power is high. What is this signal, and what causes it?
3. Why are the peaks in Fig. 1f and Fig. 1g asymmetrical about the center (in amplitude)?

The dark solid line in Fig. 3 of the revised version is calculated based on the same equation, i.e., Eq. (1), according to the figure caption. Why do the results in the revised version differ so much from the original ones in Fig. 3?

Reviewer #3 (Remarks to the Author):

The authors have addressed all my comments in this revised version. I congratulate the authors for their work and recommend their manuscript for publication in Nature Communications.

Detailed Responses to Reviewers The reviewers' comments are indicated in blue, our answers are in black.

Report 1:

The authors well addressed the points raised by the reviewers. I now recommend the publication of this paper in Nature Communications.

Answer: We thank the reviewer for recommending our manuscript for publication in Nature Communications.

Report 2:

The authors have now convinced me that the innovation presented in this work is sufficient for publication in Nature Communications, with the addition of quantitative analysis. I can now recommend it for publication, provided that the following questions are addressed:

Answer: We are very happy that the reviewer recommends publication after we have addressed four questions.

1. In Fig. 1f and 1g, why are the linewidths small when the microwave drive power is low and significantly larger when the microwave drive power is high? How is this related to the Gilbert damping of the material and the drive power?

Answer: To compare the observed linewidths with the Gilbert damping as suggested by the reviewer, we performed Lorentzian fits to the spectra indicated by the reviewer. We extracted peak positions (Fig. R1a) and Full Widths at Half Maximum (FWHM) (Fig. R1b) as a function of power. FWHM values reside between about 0.001 MHz and 10 MHz depending on the mode and power level. At low (high) power, the mode $f_p/2$ has the relatively smallest (largest) linewidths which differ clearly from the linewidth Δf expected from the Gilbert damping indicated by the horizontal line in Fig. R1b. The value of $\Delta f = 0.425$ MHz was evaluated for a realistic damping parameter $\alpha = 1 \times 10^{-4}$ for thin YIG [C. Dubs et al J. Phys. D: Appl. Phys. 50 204005 (2017)] according to $\Delta f = 2\alpha f$ with $f = 2.125$ GHz. The linewidth for $f_p/2$ is very small at power levels below 1.3 mW and considerably smaller than expected from the Gilbert damping. Here, we observe that the transverse dynamic magnetization component oscillates at $f_p/2$ when the longitudinal component oscillates at f_p . This is a coherently driven magnetic oscillation without a magnon relaxation process, allowing only a single frequency at $f_p/2$ that is not governed by Gilbert damping.

Above 2.3 mW, the linewidths of modes with $f \neq f_p/2$ are on the order of sub-MHz as the reviewer noted and much larger than the linewidth of mode $f_p/2$ at small power levels. We believe that the difference stems from different scattering processes. At high powers, the 4-magnon scattering processes set in. They allow for more magnon pairs to satisfy the energy and momentum conservation rules as stated in our Eq. 2 in the main text ($f_1 = f_p/2 - \delta$ and $f_2 = f_p/2 + \delta$). These processes are different from Gilbert damping and determined by the inhomogeneity which contributes to the broadening of dispersion, increasing the number of possible magnon pairs in frequency space. Overall the linewidths of different modes show complex dependencies on the power and a one-to-one relationship to the Gilbert damping is not found. We thank the reviewer for the question and added the evaluation of linewidths to the supplementary information.

2. I noticed a signal (peak) at the center in Fig. 1f (also in Fig. 2a at 28.6 mT) when the drive power is high. What is this signal, and what causes it?

Answer: For CPW2 we explained the high-power signal at the center of the figures by emergent branches of non-linear spin waves which approached each other and merged at $f_p/2$ with increasing power. Using Figs. 4e and 4g of the revised manuscript we had explained this convergence behavior towards $f_p/2$ with co-propagating non-linear modes which were generated by 4-magnon-scattering between CPW1 and CPW2 and resided at $k_x < 0$. As a consequence we assumed these magnons to propagate towards CPW2 (to escape from CPW1 to the right). We interpret the weak signal that the reviewer recognized such that the scattering processes happen still close enough to CPW1 (in its near field) and thereby induce a small voltage signal in CPW1. We note that one of the converging branches is weakly visible on the right side of the CPW1

Figure R1: Power-dependent **a** peak frequencies above and below $f_p/2$ and **b** their linewidths (FWHMs) including mode $f_p/2$ extracted from spectra in Fig. 1f. The straight line in **b** indicates the linewidth expected from Gilbert damping with $\alpha = 1 \times 10^{-4}$. The grey shaded area indicates the power range where multiple peaks appear, preventing fitting with a single Lorentzian function.

spectrum at 28.8 mT in Fig. 2a. We added a comment to the manuscript.

3. Why are the peaks in Fig. 1f and Fig. 1g asymmetrical about the center (in amplitude)?

Answer: We explain the asymmetry by Fig. 4a. Here, we show two magnons generated after 4-magnon scattering (blue dots) on the right side of CPW1. On the one hand, the lower frequency magnon propagates towards the right direction which is towards CPW2 so they are detected more by CPW2. On the other hand, the upper frequency one propagates towards the left so that its signal is small at CPW2. This scattering process can occur also on the left side of CPW1. Now the high-frequency mode propagates towards the right meaning that it is detected by CPW1, which is consistent with our observation that CPW1 has the stronger intensity for high frequency non-linear modes. We made a comment on this in the manuscript.

The dark solid line in Fig. 3 of the revised version is calculated on the basis of the same equation, i.e., Eq. (1), according to the figure caption. Why do the results in the revised version differ so much from the original ones in Fig. 3?

Answer: Performing the quantitative analysis of the spectra requested by the reviewer we have realized that the power-dependent data are consistently described over a broad parameter regime if we assume that the sample resided in fields which had a small constant offset of -0.75 mT (systematic error). We had specified this offset field in the Methods section. CPW2 is far away from the emitter CPW1 and the magnetization dynamics are unperturbed there. The orange curve describes the modes at CPW2 and it is based on Eq. 1 with the offset field of -0.75 mT included. At CPW1 the high applied power levels require the consideration of a power dependent frequency shift as detailed in the manuscript and reported by previous works cited in the manuscript. Taking into account both the offset field and the non-linear frequency shift due to the large driving field under CPW1 in Eq. 1, we obtain the black lines in the revised Fig. 3.

Report 3:

The authors have addressed all my comments in this revised version. I congratulate the authors for their work and recommend their manuscript for publication in Nature Communications.

Answer: We thank the reviewer for recommending the publication of our manuscript.

REVIEWERS' COMMENTS

Reviewer #2 (Remarks to the Author):

The authors have now addressed all the questions. I have no further comments and can recommend this paper for publication in Nature Communications.

Detailed Responses to Reviewers The reviewers' comments are indicated in blue, our answers are in black.

Reviewer 2:

The authors have now addressed all the questions. I have no further comments and can recommend this paper for publication in Nature Communications.

Answer: We are very happy that the reviewer recommends publication. The reviewer's critical feedback helped us to improve our manuscript.